# *Corynebacterium glutamicum* Mechanosensing: From Osmoregulation to L-Glutamate Secretion for the Avian Microbiota-Gut-Brain Axis

**DOI:** 10.3390/microorganisms9010201

**Published:** 2021-01-19

**Authors:** Yoshitaka Nakayama

**Affiliations:** 1Molecular Cardiology and Biophysics Division, Victor Chang Cardiac Research Institute, Darlinghurst, NSW 2010, Australia; y.nakayama@victorchang.edu.au; Tel.: +61-2-9295-8744; 2St Vincent’s Clinical School, Faculty of Medicine, The University of New South Wales, Darlinghurst, NSW 2010, Australia

**Keywords:** bacterial mechanosensitive channel, MscS-like channel, glutamate exporter

## Abstract

After the discovery of *Corynebacterium glutamicum* from avian feces-contaminated soil, its enigmatic L-glutamate secretion by corynebacterial MscCG-type mechanosensitive channels has been utilized for industrial monosodium glutamate production. Bacterial mechanosensitive channels are activated directly by increased membrane tension upon hypoosmotic downshock; thus; the physiological significance of the corynebacterial L-glutamate secretion has been considered as adjusting turgor pressure by releasing cytoplasmic solutes. In this review, we present information that corynebacterial mechanosensitive channels have been evolutionally specialized as carriers to secrete L-glutamate into the surrounding environment in their habitats rather than osmotic safety valves. The lipid modulation activation of MscCG channels in L-glutamate production can be explained by the “Force-From-Lipids” and “Force-From-Tethers” mechanosensing paradigms and differs significantly from mechanical activation upon hypoosmotic shock. The review also provides information on the search for evidence that *C. glutamicum* was originally a gut bacterium in the avian host with the aim of understanding the physiological roles of corynebacterial mechanosensing. *C. glutamicum* is able to secrete L-glutamate by mechanosensitive channels in the gut microbiota and help the host brain function via the microbiota–gut–brain axis.

## 1. Introduction

### Corynebacterial Mechanosensitive Channel Model for the Industrial L-Glutamate Production

The *Corynebacterium glutamicum* ATCC13032 strain was first isolated as a glutamate producer in the avian (most likely pigeon) feces-contaminated soil in 1957 by Kinoshita and Udaka [1], who described the result of an elaborated screening method with a glutamate-auxotrophic bacterium *Leuconostoc mesenteroides* strain, P-60 [2]. This sparked microbial amino acid production and our understanding *C. glutamicum* physiology has made numerous paradigm shifts in the biotechnological process. Since then, several *Corynebacterium glutamicum* subspecies, other than the ATCC13032 strain, have been isolated from different materials for industrial production. One of the most striking findings of *C. glutamicum* studies is biotin-dependent L-glutamate secretion. In nature, glutamate has two optical isomers, d(−) and l(+), and the l form (L-glutamate) is the most widely occurring, while the d form (D-glutamate) occurs rarely in the bacterial cell walls as a component of peptidoglycan layers. At a growth-limiting concentration of biotin, *C. glutamicum* massively releases L-glutamate into the culture media; however, this bacterium does not produce it at all under normal culture conditions [3]. Later, several alternative ways to trigger L-glutamate secretion have been reported, and the conditions are surprisingly diversified as impacting the cell wall and membrane, such as the adding of penicillin [4], ethambutol [5], ciprofloxacin [6], fatty acid ester surfactants [7], local anesthetics [8], and temperature upshift [9,10]. Correspondingly, a wide range of mechanistic models to explain L-glutamate secretion was suggested, and a coherent model was expected to elucidate the situation. Since L-glutamate is a charged amino acid, passive efflux was rejected, and an active carrier was proposed as the “carrier model” [11]. The leakage from damaged cell envelopes was considered as one of the main causes due to inhibited fatty acid biosynthesis and membrane alterations in L-glutamate production, proposed as the “leak model” [12,13,14]. Metabolism in the biotin-limited condition was shifted from energy production via tricarboxylic acid (TCA) cycle towards L-glutamate production. Thus, the “metabolic flow change model” was proposed to trigger L-glutamate secretion [15,16,17].

This puzzling situation has ended with the serendipitous discovery of the major L-glutamate exporter, NCgl1221 (cg1434). In the process of the screening of the L-glutamate overproducing strains without any specific treatment, several mutations on the *NCgl1221* gene were identified as the result [18]. NCgl1221 was predicted to be a homologue of bacterial mechanosensitive channels of small conductance (MscS); however, it shows similarity only in the pore domain, and its entire structure differs from other MscS-type mechanosensitive channels [19]. This finding has emphasized the importance of bacterial mechanosensing by mechanosensitive channels on the studies of corynebacterial amino acid exporters [20,21], and mechanosensitive channels are now emergent targets for transporter engineering to use *C. glutamicum* as a microbial cell factory [22,23]. Using the patch-clamp technique with *E. coli* giant spheroplasts, NCgl1221 was demonstrated to function as a mechanosensitive channel with a strong rectifying activity and slight cation selectivity. Thus, NCgl1221 was renamed as the “mechanosensitive channel of *Corynebacterium glutamicum* (MscCG)” [24]. Further, using giant protoplasts of *Bacillus subtilis,* L-glutamate was shown to be passively exported by the open pores of MscCG channels [25]. In addition to MscCG as the major L-glutamate exporter, a different mechanosensitive channel gene, *mscCG2,* was identified as a minor L-glutamate exporter in several industrial *C. glutamicum* strains, such as Z188, ATCC13869, and S9114 [26]. MscCG2 has evolved separately from MscCG, with different structural features and a different amino acid sequence; therefore, MscCG2 has been considered as a novel L-glutamate exporter. The single deletion of MscCG2 does not cause a significant decrease in L-glutamate secretion due to the presence of MscCG; however, the heterogenous overexpression of MscCG2 from the Z188 strain rescued the phenotype of the *mscCG* KO mutant of the ATCC13032 strain [26].

Conclusively, using a novel patch-clamp technique with *C. glutamicum* giant spheroplasts of the industrial strain ATCC13869, all endogenous mechanosensitive channels including MscCG and MscCG2 were recorded electrophysiologically in the native membrane environment [27]. These studies strongly supported the “mechanosensitive channel model” [28] and explained the L-glutamate secretion mechanisms as follows: (1) specific treatments alter membrane tension by inhibiting membrane lipids or cell wall synthesis; (2) MscCG and MscCG2 are activated by increased membrane tension;(3) L-glutamate is exported through the open pore of the MscCG-type mechanosensitive channels (Figure 1). In this model, mechanosensing by MscCG-type mechanosensitive channels is the central physiological phenomenon of the L-glutamate secretion. To extend our understanding of corynebacterial mechanosensing, we will summarize the current progress on studies of corynebacterial mechanosensitive channels and discuss the potential physiological significance of L-glutamate secretion in the corynebacterial habitats.

## 2. Mechanosensitive Solute Efflux System Functions for Osmoregulation, but L-Glutamate Is Not Exported in *Corynebacterium glutamicum*

Osmoregulation is an indispensable cellular function for corynebacteria to survive and adapt to fluctuating osmotic environments in their various habitats in the soil, sewage, plants, food products, animal mucosa, and skin flora [29,30,31]. The *C. glutamicum* ATCC13032 strain was discovered in avian feces-contaminated soil as a glutamate producer [32]. Other heat-tolerant glutamate producers, *Corynebacterium efficiens* and *Corynebacterium suranareeae,* were identified from onion bulb [33] and soil contaminated with starling’s feces [34], respectively. *Corynebacterium variabile* was isolated in smear-ripened cheese in a high salt environment [35,36]. In general, the cytoplasmic concentration of osmotically active solutes is higher than in the environment, thus causing water influx and cell swelling in the soil [37]. Since the cytoplasmic membrane is freely permeable to water, but forms a barrier for solutes between the environment and the cytoplasm, turgor pressure is exerted by the cytoplasmic membrane towards the cell wall. Bacterial turgor pressure has been estimated to be 3–5 atm for Gram-negative bacteria and 20 atm for Gram-positive bacteria, respectively [38]. Obviously, corynebacteria were forced to develop adaptation mechanisms to cope with a huge turgor pressure that would easily break cells if they were not protected by the cell wall. 

The main strategies of *C. glutamicum* for osmoregulation are mechanosensitive solute efflux systems after an osmotic downshift and the accumulation of compatible solutes, such as betaine, proline, glutamine, ectoine and trehalose, after an osmotic upshift [39,40,41] (Figure 2). An osmotic downshift leads to dramatically increased turgor pressure, due to an excessive water influx. To avoid osmotic cell lysis, *C. glutamicum* activates, within milliseconds, the mechanosensitive solute efflux system that consists of two types of mechanosensitive channels, MscS-type (MscCG and MscCG2) and MscL-type (CgMscL), to release intracellular solutes swiftly into the environment and reduce the driving force for water entry. The mechanosensitive solute efflux system releases betaine or proline, preferably to other amino acids of similar size, and does not excrete ATP in *C. glutamicum* [42]. This is higher substance specificity than the counterpart of the *E. coli* system that is non-selective and releases even small proteins. After hypoosmotic shock, the cytoplasmic concentration of L-glutamate does not change even after glutamate synthesis after K^+^ accumulation in a hyperosmotic condition [43,44]. Thus, it is questionable whether L-glutamate plays an important role as an osmolyte in the osmoregulation in *C. glutamicum*.

The bacterial mechanosensitive solute efflux system has been studied comprehensively by characterizing *E. coli* mechanosensitive channels of large conductance (MscL) and of small conductance (MscS) [45]. *E. coli* has one *MscL* gene and six *MscS* genes (MscS, YnaI, YbdG, YbiO, YjeP(MscM), KefA(MscK)) [46,47]. *E. coli* MscL and MscS mechanosensitive channels are required for survival upon hypoosmotic shock, and the double knockout among seven mechanosensitive channel genes leads to an almost 90% decrease in survival rate in *E. coli* [48]. In contrast, the *C. glutamicum* ATCC13032 strain has one MscL-like channel (CgMscL) and only one MscS-like channel (MscCG). Interestingly, several industrial strains, such as Z188, ATCC13869, and S9114, have two structurally different MscS-like channels (MscCG and MscCG2). Nottebrock et al. demonstrated, with the *C. glutamicum* ATCC13032 strain, that the deletion of all mechanosensitive channel genes *CgMscL* and *MscCG* did not change survival rate after osmotic downshock [49]. This shows that the corynebacterial mechanosensitive solute efflux system is not required for survival or that unidentified osmoresponsive channels further exist for hypoosmotic response. Instead of functioning as osmotic safety valves, corynebacterial mechanosensitive channels play roles in the fine-tuning of cytoplasmic osmolarity in the hyperosmotic environment. The mechanosensitive channel MscCG is coupled with the activity of betaine transporter BetP to regulate the cytoplasmic betaine concentration as the “pump and leak model”, that MscCG exports and BetP imports betaine to balance cytoplasmic osmolarity [24] (Figure 2). This indicates that the corynebacterial mechanosensitive efflux system exports mostly betaine as an osmotically active solute rather than L-glutamate for osmoregulation. 

## 3. Diversity of MscS Mechanosensitive Channel Superfamily and the Impact of the MscCG Channel Gating on L-Glutamate Secretion

The existence of MscS-type mechanosensitive channels in the cytoplasmic membranes of both prokaryotic organisms, *E. coli* and *C. glutamicum,* indicates that these channels have developed through evolution to cope with changes of mechanical environments; however, its functions as an osmoregulator are significantly different [50]. *E. coli* MscS has a large open channel pore (~16 Å), and thus its conductance reaches 1 nS, whereas *C. glutamicum* MscCG has a significantly smaller conductance of approximately 0.3 nS [24,27,51]. As an osmoregulator, *E. coli* MscS has strong inactivation and adaptation mechanisms to mechanical stimuli that are needed to avoid the over-efflux of cytoplasmic molecules upon osmotic downshock [52,53]. However, *C. glutamicum* MscCG does not have these features and tends to be open as a metabolic valve rather than an osmotic safety valve [54]. Recent studies of the MscS channel superfamily revealed that MscS-like channels are present among cell-walled organisms, bacteria, archaea, fungi, algae, and plants, and the physiological functions of these MscS-like channels are not simply those of osmotic safety valves upon hypoosmotic shock [19]. In eukaryotes, algal and plant MscS-like channels (MSCs, MSLs) are much more complicated structures than *E. coli* MscS and are found in the organellar membranes of chloroplasts [55,56,57] and mitochondria [58,59,60] for mechanosensing. Fungal MscS-like channels (Msy1 and Msy2) are localized in the endoplasmic reticulum membranes and are involved in osmotic Ca^2+^ signalling upon hypoosmotic shock [61,62]. 

Although the physiological functions of corynebacterial MscCG-type mechanosensitive channels are still controversial, L-glutamate secretion is certainly caused by the conformational changes of the MscCG and MscCG2 channels. *C. glutamicum* MscCG and MscCG2 are characterized to have the N-terminal pore domain (1–286 aa), corresponding to the entire *E. coli* MscS that has three transmembrane helices (TM1, TM2, and TM3) and a cytoplasmic cage domain (Figure 3A). Recently, Reddy et al. refined the 3D structures of the full-length *E. coli* MscS embedded in lipid bilayers by cryo-electron microscopy with nanodiscs [63] (Figure 3B). MscCG and MscCG2 show the highest similarity in the pore-forming helix TM3 and adjacent regions. The unique feature of the mechanosensitive channel MscCG is the C-terminal domain (287–533 aa), including a cytoplasmic loop, the fourth transmembrane helix, TM4, and a periplasmic loop. In contrast, MscCG2 does not have these features (Figure 3C). This structure is highly conserved only in corynebacteria. Several gain- and loss-of-function mutations on MscCG were identified by the glutamate productivity assay and bacterial patch-clamp technique (Figure 3C). Originally, Nakamura et al. identified W15CSLW, A100T, A111T, A111V in the N-terminal pore domain and V419::IS1207, P424L in the periplasmic loop during the screening of L-glutamate overproducing strains. These mutations were reported to cause a spontaneous L-glutamate secretion in *C. glutamicum* [18]. Afterwards, Nakayama et al. reported using *E. coli* patch-clamp that the spontaneous L-glutamate secretion was caused by the gain-of-function mutation on mechanosensitivity of MscCG channels [64]. These findings proved that MscCG mechanosensitivity can be evaluated by L-glutamate productivity in *C. glutamicum.* Using these assays, the functional domain of MscCG has been thoroughly investigated. Yamashita et al. reported, using L-glutamate productivity assay, that the N-terminal pore domain (1–286 aa) is essential to export L-glutamate, and suggested, using homology modelling, that MscCG has an extra small loop structure (221–232 aa) and its deletion resulted in the loss of channel functionality [65]. Becker et al. reported additional point mutations in the N-terminal pore domain and the impact of the C-terminal domain on the channel function [66,67]. A106V was the gain-of-function mutation to cause the spontaneous L-glutamate secretion, and interestingly, Q112L and V115S double point mutation caused a loss-of-function mutation, such that *C. glutamicum* cannot export L-glutamate, even with penicillin treatment. In contrast, the deletion of the periplasmic loop (423–533 aa) in the C-terminal domain caused spontaneous L-glutamate secretion, but not the further deletion of the periplasmic loop and the fourth transmembrane helix TM4, indicating that the fourth transmembrane helix TM4 is involved in the mechanosensitivity of MscCG channels. Moreover, Krumbach et al. further identified the gain-of-function mutations V422K, V422D, E423P, and E423S using CRSPER/Cas12a genome-editing technology [68]. Based on the position of these mutations, the interaction with other components of the cell wall was suggested to be crucial for MscCG mechanosensitivity. 

## 4. “Force-From-Lipids/Tethers” Paradigms and Bacterial Cell/Membrane Mechanics for Mechanosensing

In understanding mechanosensing by MscCG-type mechanosensitive channels, importantly, hydrostatic pressure itself, such as osmotic pressure, does not directly activate force-sensing ion channels. Martinac et al. first demonstrated the existence of MscS channels as “pressure-sensitive” channels in *E. coli* giant spheroplasts, since suction was applied to activate the channels [69]. Later, the responsible gene, *yggB*, was identified, and the purified protein was successfully reconstituted into liposomes to confirm its mechanosensitivity in lipid bilayers [70]. Using a high-speed camera, the membrane deformed by suction was visualized to calculate membrane tension [71], and the activation threshold of MscS channels was estimated to approximately 6 mN/m [72]. Moreover, Martinac et al. elaborately proved that amphipaths, such as Lysophosphatidylcholine (LPC) and chlorpromazine (CPZ), which generate local membrane curvature, can activate *E. coli* mechanosensitive channels MscS without applying suction [73]. These findings led to establishing the concept of the “Force-From-Lipids” paradigm for mechanosensing, that the membrane protein always senses the transbilayer force profile transmitted through the membrane lipids [74,75,76] (Figure 4). In this paradigm, mechanical stimuli, deforming cell shape, such as pressure, contact, and indentation, increase membrane tension at the cellular level (micrometer-scale), but more importantly, change in the transbilayer force profile at the molecular level (nanometer scale) [77,78]. For individual mechanosensitive channels in the cytoplasmic membrane, global membrane curvature on the micrometer-scale is too large to sense force. For instance, we experience on the Earth while being unable to feel its curvature on the ground. Thus, nanometer-scale force-sensing is more critical for the activation of bacterial mechanosensitive channels than on micrometer-scale [73,79]. In contrast to the “Force-From-Lipids” paradigm, most of the eucaryotic mechanosensitive channels (TRPs and PIEZOs) are connected to the cytoskeleton as molecular tethers, and mechanical force is transmitted directly or indirectly through the tethers to activate the channels [80,81,82]. This idea is also widely accepted as the “Force-From-Tethers” paradigm (Figure 4). For corynebacterial mechanosensing by MscCG-type mechanosensitive channels, the cell wall may function as extracellular molecular tethers to transmit the mechanical force to the channels by connecting to the extracellular loop of MscCG channels since the gain-of-function mutations that cause spontaneous L-glutamate secretion are localized at the extracellular boundary [68]; however, an open question is why MscCG2 channels are also activated by penicillin treatments without the extracellular loop [26].

The mechanical properties of bacterial cells and membranes are distinct from animal counterparts since bacterial cells are protected by the cell wall and, thus, are up to several hundred thousand times stiffer than animal cells [83,84]. Young’s elastic modulus, shear modulus, bending stiffness, and viscosity, are important mechanical properties of biological cell membranes. For the Gram-negative and Gram-positive bacterial cells, Young’s elastic modulus was estimated between 50–150 and 100–200 MPa, respectively [83]. Corynebacteria have a mycomembrane, including the mycolic acid layer, which is thicker than the cell wall of the Gram-positive bacteria, and thus this value is even larger. Based on these properties, the bacterial cell wall sustains approximately 90% of turgor pressure and leaves a 10% share to be sustained by the cytoplasmic membrane [85]. This “force sharing” between the cell wall and membrane is large enough to activate mechanosensitive channels activated by membrane tension in the range of 6–12 mN/m in the cell membrane. In addition to bacterial cell mechanics, bacterial membrane mechanics are difficult to study due to the bacterial cell size and the cell wall. Using the micropipette aspiration technique, the mechanical membrane properties of *C. glutamicum* giant spheroplasts were evaluated to be much softer compared to the *E. coli* giant spheroplast membranes [27]. *C. glutamicum* has a significant number of the negatively charged lipids cardiolipin (CL), phosphatidylglycerol (PG), and phosphatidylinositol (PI), compared to phosphatidylethanolamine (PE) in the other bacterial cytoplasmic membrane. Consistently, liposomes made of the negatively charged lipids, DOPG, were shown to have softer mechanical properties than liposomes made of the neutral lipids DOPC by atomic force microscopy [86]. These findings indicate that corynebacterial membranes are much more deformable by mechanical stimuli than other bacterial membranes, and, therefore, that membrane tension, increased by largely expanded membranes upon osmotic downshock, will activate *C. glutamicum* mechanosensitive channels, MscCG, MscCG2 and CgMscL in the cytoplasmic membranes.

## 5. Mechanosensitive Channel Activation by Altering the Cell Membrane and Wall with Specific Triggers

In the mechanosensitive channel model, it has been proposed that membrane tension increase, caused by specific triggers, activates MscCG-type mechanosensitive channels for L-glutamate secretion [23,28]. In truth, MscCG-type mechanosensitive channels can be activated mechanically by increasing membrane tension with suction [27]; however, the activation mechanisms in L-glutamate production are still mysterious. Firstly, the activation of MscCG-type mechanosensitive channels is exclusive in the presence of the other type of mechanosensitive channel MscL. Secondly, the cell shape and size are not dramatically changed; thus, it is unlikely that membrane tension is increased as during osmotic cell expansion. Thirdly, the conserved C-terminal domain of MscCG channel has impacts on L-glutamate secretion, although its molecular function is not understood. This indicates that MscCG-type mechanosensitive channels sense force from the lipids in the cell membrane in L-glutamate production differently from osmotic swelling. 

The currently reported specific triggers to induce L-glutamate production can be grouped, such as the inhibition of fatty acid biosynthesis, the alteration of mechanical properties of the cell membrane, and the degradation of the cell wall (Figure 5). The *C. glutamicum* fatty acid biosynthesis system requires biotin, since the α subunit of acetyl CoA carboxylase complex, AccBC, is biotinylated [87]. Biotin limitation decreases the catalytic reaction to synthesize malonyl CoA from acetyl CoA and to synthesize α-carboxyl-acyl-CoA from acyl-CoA; thus, the following fatty acid biosynthesis is inhibited as resulting in the alteration of both membranes, the cell membrane and the mycolic acid layer (Figure 5). Other than biotin limitation, the fatty acid ester surfactants, Tween 40 (Polyoxyethylene sorbitan monopalmitate) and Tween 60 (Polyethylene sorbitan monostearate), induce L-glutamate production, but Tween 20 (Polyoxyethylene sorbitan monolaurate) and Tween 80 (Polyoxyethylene sorbitan monooleate) do not. To understand the molecular mechanisms, the *dtsR* (detergent sensitivity rescuer) gene was isolated as a multicopy suppressor of a Tween 40-sensitive mutation of *C. glutamicum* that requires fatty acid for growth and secretes L-glutamate without biotin limitation [14]. DtsR is the β subunits of the acetyl CoA carboxylase complex and renamed later AccD1 since *C. glutamicum* has four acetyl CoA carboxylase β subunit genes (AccD1–4). AccD1 is involved in the catalytic reaction to synthesize malonyl CoA from acetyl CoA with AccBC (α-subunit) and AccE (ε-subunit), whereas AccD2 and AccD3 are involved in mycolic acid synthesis. Adding Tween 40 decreases the expression level of AccD1, and thus results in the inhibition of fatty acid biosynthesis to induce L-glutamate secretion, as with biotin limitation. From malonyl-CoA, *C. glutamicum* synthesizes fatty acids with the type-I fatty acid synthesis system, which consists of two types of fatty acid synthase, FasA and FasB [88]. Unlike other bacteria, *C. glutamicum* does not have a type-II fatty acid synthesis system. FasA is the main synthase for synthesizing oleoyl-CoA (C18:1) and palmitoyl-CoA (C16:0), and the deletion of FasA is lethal, whereas FasB is the subordinate synthase for saturated stearoyl-CoA (C18:0) and palmitoyl-CoA (C16:0). The *fasA* mutant requires oleic acids (C18:1) for growth and spontaneously secretes L-glutamate, implying that oleic acid auxotrophy is involved in the activation of MscCG mechanosensitive channels by changing membrane lipids. Furthermore, in fatty acid biosynthesis, *C. glutamicum* synthesizes phospholipids from phosphatidic acid (PA) for making cell membranes that consist of negatively charged phospholipids: phosphatidylglycerol (PG), cardiolipin (CL), phosphatidylinositol (PI), and phosphatidylinositol mannosides (PIMs). The gene expression for each phospholipid synthesis alters the membrane lipid components. The overexpression of cardiolipin synthase *cls* causes spontaneous glutamate secretion without any treatment [89], implying that the increased amount of cardiolipin in the cell membrane may activate MscCG mechanosensitive channels for glutamate efflux. Cardiolipin is a four-tailed phospholipid, and its structure resembles two phosphatidylglycerols joined via the head groups. Due to its inversed conical shape, cardiolipin is a non-bilayer lipid and contributes to creating membrane curvature, such as the curved poles of rod-shaped bacteria [90]. 

The mechanical properties of cell membranes affect mechanosensing because the force is transmitted to mechanosensitive channels through the viscoelastic cell membrane [91]. Alterations in the membrane-lipid composition, the ratio of saturated/unsaturated lipids, and the membrane lipid chain length, have been reported to change the mechanosensitivity of *E. coli* MscS and MscL, as shown in modelling studies [72,92,93]. In *C. glutamicum*, a temperature upshift from 30 °C to 37–41 °C induces L-glutamate secretion, and this method has been considered as one of the most cost-effective methods of industrial production [9,94,95]. The cell envelope fluidity of *C. glutamicum* was analyzed with 1-(4-trimethylammoniumphenyl)-6-phenyl-1,3,5-hexatriene (TMA-DPH) during temperature-triggered glutamate production, and a significant increase in fluidity was reported [96]. The membrane fluidity increase is not caused by the high temperature itself, but mainly by changing membrane lipid composition after temperature upshift. This indicates the possibility of the temperature-dependent activation mechanisms of MscCG channels by changing membrane lipid composition. Related to temperature-triggered glutamate production, Lambert et al. reported that local anesthetics, such as chlorpromazine, tetracaine, butacaine, and benzocaine, also trigger L-glutamate secretion by modulating the membrane state [8]. In this method, the viscosity or fluidity of the membrane was proved to be significantly unchanged using electron spin resonance spectroscopy with spin-labelled fatty acid probes. Thus, it was suggested that lipid bilayer elasticity, membrane shape, and membrane bending energy, are critical, as opposed to membrane viscosity and fluidity. Since local anesthetics, such as amphipaths, can be attributed to creating local membrane curvature, to change the transbilayer force profile [73], the activation of MscCG channels by local anesthetics can be explained by the “Force-From-Lipids” mechanosensing paradigm (Figure 5). Supportive of the above idea, the effects of local anesthetics and osmotic shifts were mutually interchangeable. A hyperosmotic shift inhibited tetracaine-triggered glutamate efflux, whereas a hypoosmotic shift enhanced the action of tetracaine [8].

The corynebacterial cell wall has a unique cell surface structure only seen in the Corynebacteria–Mycobacteria–Nocardia group [97]. The mycolic acid layer, as the outer membrane, mainly consists of trehalose corynomycolate [98]. Underneath the mycolic acid layer, the covalently linked arabinogalactan layer and peptidoglycan layer are present between the mycolic acid layer and cell membrane. Ethambutol and penicillin inhibit a series of arabinosyltransferases and enzymes for the cross-linking of peptidoglycans, respectively. Thus, these antibiotics cause the weakened cell wall due to a lower amount of arabinogalactan and fewer cross-links of the peptidoglycans. As a specific trigger, adding ethambutol or penicillin induces L-glutamate secretion, such that MscCG-type mechanosensitive channels are activated by a strongly disordered cell envelope due to the degradation of the cell wall structure. To establish an efficient cellulosic glutamate production from lignocellulose feedstocks, the activation of MscCG by ethambutol and penicillin was targeted [20]. Without the mechanical support of the cell wall, cells are easily expanded by turgor pressure, which may cause the cells to burst; thus, they need to immediately activate the mechanosensitive solute efflux system. However, L-glutamate secretion, caused by adding ethambutol and penicillin, is caused by the activation of MscCG-type channels exclusively, similar to biotin limitation and adding fatty acid surfactants rather than osmotic swelling, which activates all mechanosensitive channels. The DNA microarray assay revealed that, after adding ethambutol or penicillin, the expression of MscCG is significantly increased [5], indicating that the functions of MscCG channels are involved in the cell wall integrity. Other than ethambutol and penicillin, the effects of ciprofloxacin were proposed to change the cell wall integrity as well. Ciprofloxacin is an antibiotic to treat bacterial infection and inhibits a type II topoisomerase (DNA gyrase) that is necessary to separate DNA, thereby impairing bacterial cell division. DNA gyrase activity is linked to peptidoglycan biosynthesis because glutamate racemases, which convert L-glutamate to D-glutamate, are known to inhibit DNA gyrase activity, and D-glutamate is present in peptidoglycan cross-links [99,100,101]. Thus, ciprofloxacin alters cell wall integrity to trigger L-glutamate production; however, unlike ethambutol and penicillin, ciprofloxacin-triggered L-glutamate production is independent of MscCG activity and needs to be studied further. Based on the studies of the structural alterations of the cell wall by ethambutol and penicillin, it can be explained that MscCG channels are activated by force transmitted to the channels tethered by peptidoglycan as “Force-From-Tethers” mechanosensing (Figure 5). 

## 6. Soil and Gut Bacterium Scenario: L-Glutamate Secretion for the Environmental Signals in the Soil and the Microbiota-Gut-Brain Axis in the Gut

*C. glutamicum* has been considered as Gram-positive “soil” bacterium, and its L-glutamate export is for the “excretion” to reduce turgor pressure to protect cells from osmotic lysis. However, corynebacterial mechanobiology studies do not support this idea. The corynebacterial mechanosensitive solute efflux system is mainly adjusting the cytoplasmic concentration of betaine, and L-glutamate is not exported to reduce cytoplasmic osmolarity. Notably, mechanosensitive channels are not required for survival, although turgor pressure upon osmotic downshock activates all mechanosensitive channels by increased membrane tension. In the soil environment, biotin-limiting exposure to surfactants and penicillin to trigger L-glutamate production can happen with other microbes; however, it is unlikely that *C. glutamicum* excretes L-glutamate for adapting to osmotic environments. This raises the question: What is the physiological significance of the L-glutamate production for *C. glutamicum*?

Recent studies have revealed that L-glutamate is used for living organisms as a signaling molecule rather than an osmolyte. In soil environments, plants sense the gradient of L-glutamate concentration with roots for nutrient foraging and shaping root architecture. Plant root architecture is genetically determined; however, it is highly flexible to adapt to soil environments and the chemical and mechanical characteristics of the soils ultimately shape the architecture by changing root growth [102,103]. Plants adopt a “foraging strategy” with lateral root proliferation to enhance nutrient access; thus, they always try to change root development towards nitrogen (N) sources. Amino acids are omnipresent as a nitrogen source in the soil environment at low concentrations. Among the 20 proteinogenic amino acids, L-glutamate was demonstrated to cause the most intense responses in the roots of *Arabidopsis thaliana* [104]. It should be noted that L-glutamate acts as a modifier for root growth and branching, similar to the well-known plant-growth regulator auxin (indole-3-acetic acid, IAA) [105]. Plant cells exposed to L-glutamate elicit rapid membrane depolarization and a transient increase in [Ca^2+^]_cytosol_ [106,107]. This is clear evidence that L-glutamate creates electrical signals in plants, as observed in mammalian synaptic transmission [108]. L-glutamate is common in phloem sap and propagates signals over a long distance even between the root and shoot [109]. This enables plants to regulate the nitrogen/carbon status by adjusting the nitrate uptake system from the soil [110]. Moreover, L-glutamate is used as a signal for bacterial cell–cell communications. Bacterial cells communicate through electrical signaling with ion channels to share information about surrounding environments [111]. Bacterial biofilms expand outward until cells in the interior consume the available reserves of L-glutamate. The biofilms stop expanding when the L-glutamate concentration is low; however, they expand again when L-glutamate is replenished [111]. Based on these findings, it is obvious that L-glutamate is one of the environmental signals for plants and microbes in soil environments. As a soil bacterium, *C. glutamicum* may secrete L-glutamate into the surroundings to communicate and interact with plants and other microbes (Figure 6). Recent root microbiome studies revealed that the plant root exudation of metabolites, including L-glutamate, is an important mediator to interact with soil microbes [112]. Plant roots and microbes in the root vicinity (root microbiome) form highly complex ecosystems called the “rhizosphere”. The concentration of L-glutamate at the rhizosphere is several-fold higher than the bulk soil that is generally lower than 10 μM. Although the detection of corynebacteria from microbiota in the rhizosphere is rare [113] and, even in the bulk soil, actinobacteria is a minor population [114], L-glutamate produced by *C. glutamicum* can be a source of L-glutamate in the soil for environmental signals. 

In addition to roles as a soil bacterium, it should be emphasized that the *C. glutamicum* ATCC13032 strain was originally isolated in avian (most likely pigeon) feces-contaminated soil in a Japanese park. Thus, *C. glutamicum* can be a “gut” bacterium in avian animals. Birds are globally distributed with extreme morphological and ecological diversity, and the bird lineage is approximately 150 million years old with regard to symbiotic relationships with microorganisms. Over 10,000 species of birds vary in life-history traits, such as migratory behavior, flight capacity, diet, mating systems, longevity, and, thus, their gut microbiota are significantly different from other animals. Recent avian gut microbiome studies have revealed the evolutionary history and ecological significance of host-associated microbiota. Comparative gut microbiota from 59 neotropical bird species in Costa Rica and Peru were investigated because birds in neotropics are the most diverse on the earth. As a result, four bacterial phyla were detected: Proteobacteria, Firmicutes, Bacteroidetes and Actinobacteria, comprising an average of 46.3%, 37.3%, 3.3%, and 1.4%, respectively [115]. Supportively, a review of current knowledge about the avian gut microbiota also suggested that actinobacteria are the fourth most abundant phylum of microbes in the wild bird gastrointestinal tract [116]. Since birds are extremely diverse and their gut microbiome can be changed by many factors, an important question is whether pigeons have or used to have *Corynebacterium glutamicum* in their gut. Previously, actinobacteria, such as *Corynebacterium*, *Propionibacterium* and *Streptococcus*, were known as laboratory contamination if the sample contains low microbial biomass because these bacteria are common human skin-associated organisms [117], thus the detection of actinobacteria has been carefully discussed. Recent probiotics studies with pigeons revealed that pigeon milk microbiota, transmitted from parent pigeons to squabs, contain a considerable proportion of *Lactobacillus* and *Bifidobacterium,* but not *Corynebacterium* [118]. However, the dramatic effects of probiotics on microbiome profiling were demonstrated in Birmingham Roller pigeons [119]. Birmingham Roller pigeons have been domesticated for pigeon fancying due to the ability to perform backwards somersaults during flight. This flight display is the main activity for competitive Birmingham Roller shows, and, recently, the effectiveness of probiotics in domestic pigeons was tested to ascertain if pigeon health and fitness was improved for the Rollers’ flight performance. Interestingly, the pigeon gut microbiome is highly dependent on diet. When pigeons are fed only a diet of grain, actinobacteria is the most dominant phyla (more than 40% on average), and *Corynebacterium* is significantly abundant; however, when a probiotic diet was given, immediately a decrease in two genera (*Corynebacterium* and *Peptostreptococcus*) occurred, and the gut microbiome shifted towards a Firmicutes-dominated population, mainly by *Lactobacillus,* ranging from 71.8% to 99.9% in 14 days [119]. It was concluded that *Corynebacterium* and *Peptostreptococcus* are pathogens in the pigeon gut; therefore, reducing pathogen loads by probiotics helps the pigeon health. However, it should be noted that not all *Corynebacterium* species are pathogenic, *Corynebacterium glutamicum* is a non-pathogenic species and may help the host health by supplying L-glutamate in the gut. Although domestic Birmingham Roller pigeons in England and wild pigeons in Japan in the 1950s are not comparable, due to completely different diets and geographic distance, this finding strongly suggests that the *C. glutamicum* ATCC13032 strain was isolated from pigeon feces rather than from soils. The gut microbiome is broadly involved in the digestion of food products and helps nutritional uptake for the host. For host physiology, L-glutamate is absorbed by colonocytes and transferred from the lumen to portal circulation (Figure 6). In the brain, L-glutamate is the major excitatory neurotransmitter for N-methyl-d-aspartate (NMDA) receptor-mediated glutamatergic signaling, and thus the microbiota–gut–brain axis affects the cognitive function of the host [120]. Moreover, Chang et al. reported that plasma D-glutamate levels are associated with cognitive impairment in Alzheimer’s disease, and suggested that L-glutamate, produced by bacteria, such as corynebacteria in the gut, may modulate glutamatergic signaling in the brain, since L-glutamate can be converted into the D-form [121]. Moreover, L-glutamate is a substrate for gamma-aminobutyric acid (GABA), produced by *Lactobacillus* and *Bifidobacterium* in the gut, and it changes the host’s emotional behavior and moods on the microbiota–gut–brain axis on animal behaviors [122]. Although avian gut microbiota is different from humans, it is likely that *C. glutamicum* secretes L-glutamate for the brain function of the avian host. The MscCG mechanosensitive channel can be activated by the animal’s body temperature to secrete glutamate, especially in birds, because bird body temperature is the highest among animals and reaches 41–43 °C [123]. The “soil” and “gut” bacterium scenario for the habitats of *C. glutamicum* should be reconsidered to understand *C. glutamicum* physiology for the L-glutamate secretion. 

## 7. Conclusions and Prospects

Mechanosensing is ubiquitously present for osmoregulation in bacteria. To manage huge turgor pressure, most of the bacteria excrete L-glutamate to reduce the osmotic gradient. However, *C. glutamicum* L-glutamate secretion is not simply for osmoregulation, although MscCG-type mechanosensitive channels are activated to regulate cytoplasmic osmolarity. It is worth mentioning that corynebacterial mechanosensitive channels are not required for survival upon hypoosmotic shock at all, due to its cell and membrane mechanics. The activation mechanism of MscCG-type mechanosensitive channels by altering cell membrane and wall in L-glutamate production differs significantly from the osmotic activation. However, the “Force-From-Lipids” and “Force-From-Tethers” paradigms for the gating mechanisms of mechanosensitive channels can explain the lipid modulation activation of MscCG channels in L-glutamate production. *C. glutamicum* likely uses L-glutamate as environmental signals to interact with other microbes and plants as a soil bacterium. Moreover, *C. glutamicum* can also be a gut bacterium to produce glutamate for the host brain function by the microbiota–gut–brain axis. Glutamate-producing corynebacteria have been found in the avian feces-contaminated soil and, therefore, the physiological significance of *C. glutamicum* L-glutamate secretion should be reconsidered in both soil and gut bacterium scenarios. In summary, understanding corynebacterial mechanosensing mechanisms by mechanosensitive channels promises to shed light on the physiological significance of *C. glutamicum* L-glutamate secretion and to contribute to establishing sustainable developments for amino acid production by transporter engineering.

## Figures and Tables

**Figure 1 microorganisms-09-00201-f001:**
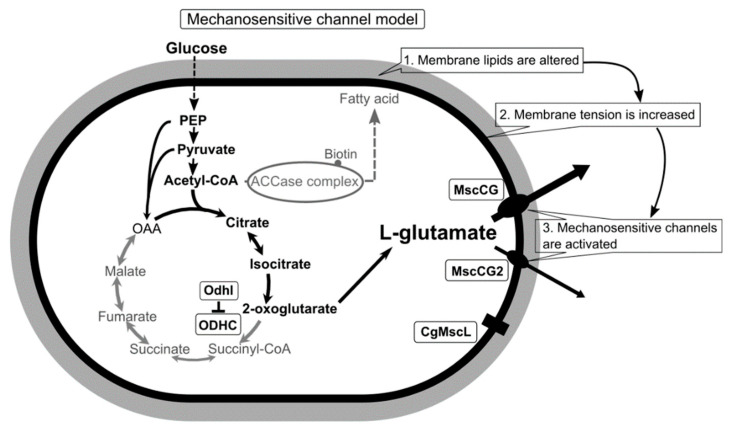
A scheme of the *C. glutamicum* L-glutamate secretion triggered by biotin limitation as mechanosensitive channel model. Biotin limitation shifts metabolic flow to produce L-glutamate by inhibiting the 2-oxoglutarate dehydrogenase complex (ODHC) activity and inhibits the acetyl CoA carboxylase (ACCase) complex activity. As a result of fatty acid biosynthesis inhibition, membrane lipids are altered to increase membrane tension. In the ATCC13032 strain, the MscS-like mechanosensitive channel MscCG is activated exclusively as major exporter by increased membrane tension to release L-glutamate. In the industrial strains Z188, ATCC13869, and S9114, another MscS-like mechanosensitive channel, MscCG2, is also activated as a minor exporter. However, the MscL-type mechanosensitive channel CgMscL is not activated in L-glutamate secretion.

**Figure 2 microorganisms-09-00201-f002:**
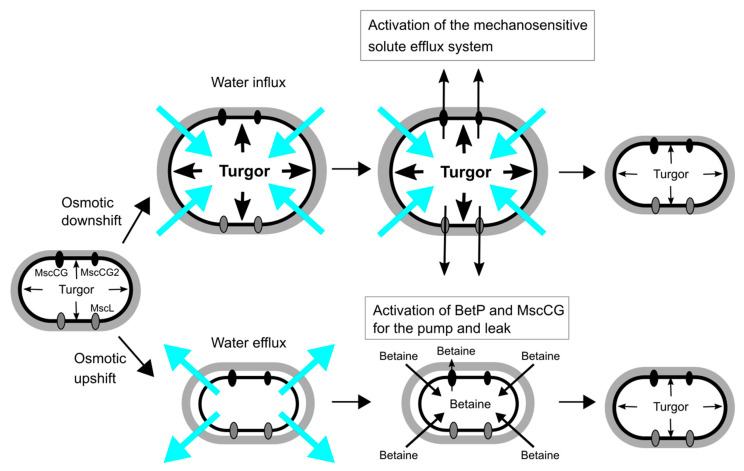
Osmoregulation of *C. glutamicum* for osmotic down-/up-shift. Turgor pressure is increased by water influx upon osmotic downshift, and the mechanosensitive solute efflux system, which consists of the mechanosensitive channels, MscCG, MscCG2 and CgMscL, is activated to reduce the osmotic gradient within milliseconds. Osmotic upshift causes water efflux, and thus cells intake betaine as major osmolytes from the environment by the activation of the betaine transporter BetP. The cytoplasmic betaine concentration is fine-tuned by the leakage through the mechanosensitive channel MscCG as the pump and leak mechanism.

**Figure 3 microorganisms-09-00201-f003:**
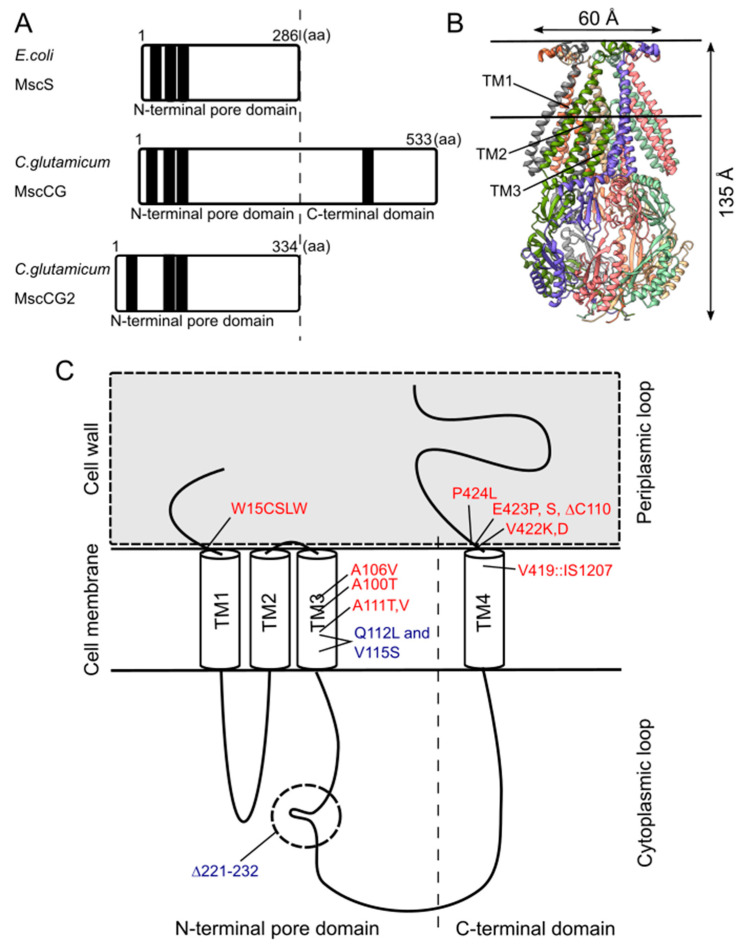
Structural features of the mechanosensitive channel MscCG. (**A**). Secondary structure comparison among *E. coli* MscS, *C. glutamicum* MscCG, and MscCG2. Black bars show predicted transmembrane (TM) helices by TOPCONS program (https://topcons.net/). (**B**). The cryoEM 3D structure of *E. coli* MscS embedded in nanodiscs with POPC:POPG = 1:4 (structure was cited from the Protein data bank 6PWP). (**C**). The domain structure of the MscCG channel and the gain (**red**)-and loss(**blue**)-of-function mutations.

**Figure 4 microorganisms-09-00201-f004:**
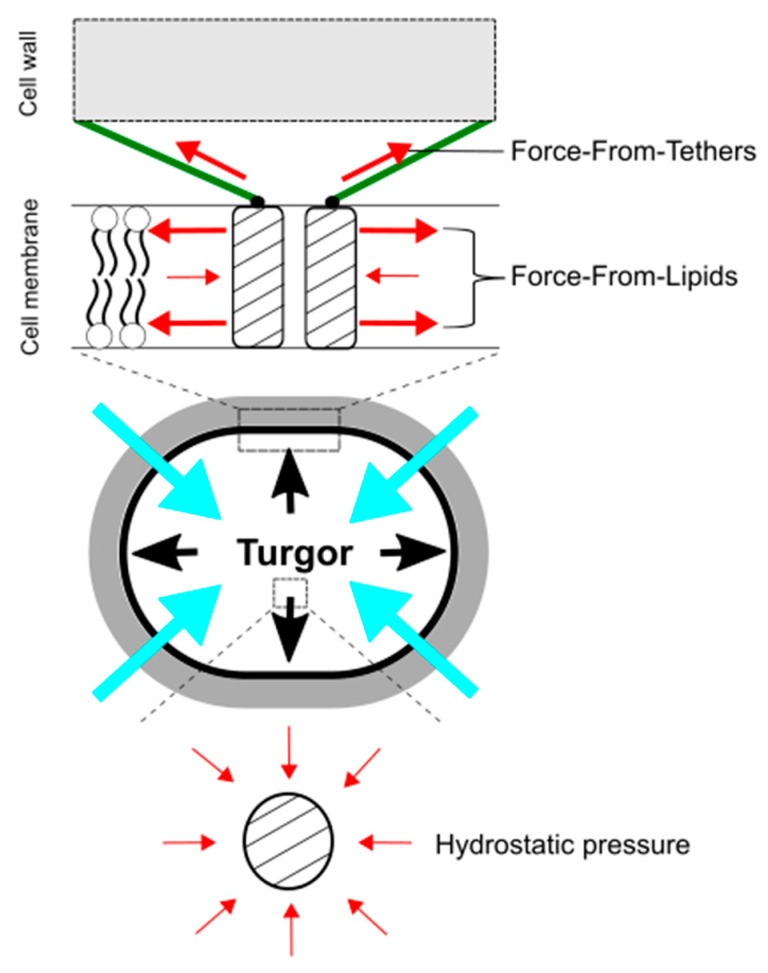
Force-From-Lipids and Force-From-Tethers paradigms for mechanosensing by mechanosensitive channels upon hypoosmotic downshift. Mechanosensitive channels sense transbilayer force profiles in the lipid bilayers and are activated by increased membrane tension. Molecular tethers, such as cytoskeleton and peptidoglycan, also transmit mechanical force to activate mechanosensitive channels (**top**). In contrast, hydrostatic pressure does not directly activate mechanosensitive channels (**bottom**). The red, black, and blue, arrows show force to ion channels, turgor pressure, and water influx, respectively.

**Figure 5 microorganisms-09-00201-f005:**
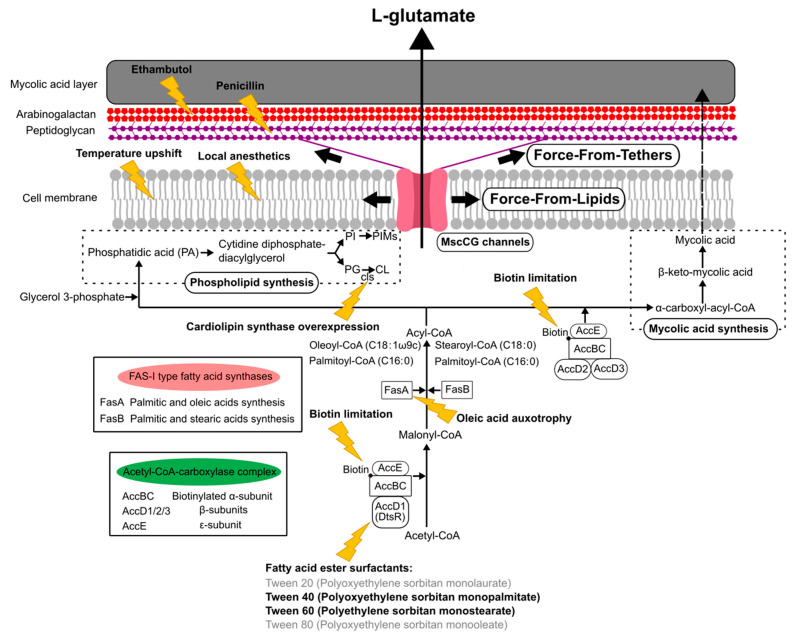
A scheme of “Force-From-Lipids/Tethers” activation mechanisms of the mechanosensitive channel MscCG in industrial L-glutamate production. Biotin limitation, adding fatty ester surfactants, oleic acid auxotrophy, and overexpression of cardiolipin synthase, alter cell membranes and mycolic acid layers by changing fatty acid, phospholipid, and mycolic acid biosynthesis. Temperature upshifts and local anesthetics change membrane mechanical properties. Adding ethambutol and penicillin degrades arabinogalactan and peptidoglycan layers in the cell wall, respectively, and activates MscCG channels.

**Figure 6 microorganisms-09-00201-f006:**
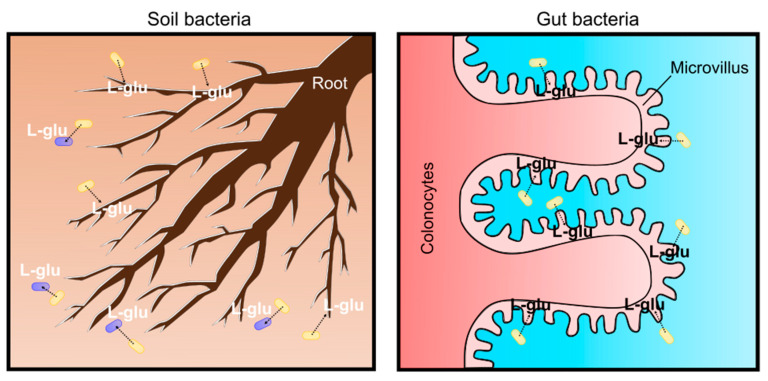
Physiological significance as soil and gut bacterium for *C. glutamicum* L-glutamate secretion. In the soil, L-glutamate is an environmental signal to communicate and interact with plants and microbes (**left**). In the avian gut, L-glutamate produced by bacteria is absorbed in colonocytes and circulated for the microbiota–gut–brain axis (**right**).

## Data Availability

The data presented in this study are available upon request from the corresponding author.

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
