# Peer review of "Corynebacterium glutamicum Mechanosensing: From Osmoregulation to L-Glutamate Secretion for the Avian Microbiota-Gut-Brain Axis"

_microorganisms, 2021, doi:10.3390/microorganisms9010201_

Round 1

Reviewer 1 Report

The review "Corynebacterium glutamicum mechanosensing: From osmoregulation to L-glutamate secretion for the avian microbiota-gut-brain axis" gives an update about the scientific knowledge about glutamate secretion. The review discusses the origin of C.glutamicum as a soil bacterium and as a hypothetic gut bacterium from avian.

I recommend the publication after minor revisions:

  • the "L" in L-glutamate should be written in an appropriate way
  • line 47: please rephrase "but not in the normal culture"
  • line 51: please also refer to ciprofloxacin to trigger glutamate production (doi: 10.1186/s12866-016-0857-6)
  • line 164: please make clear wheater Cg possesses one or two MscS-like genes
  • line 400: please replace "with" by "of" (consists of)
  • I would ask the author to discuss the effect of adding ciprofloxacin for improved glutamate secretion in more detail
  • What does a gut microbiome in avian usually look like? Are there data published which support the idea of Cg as a gut microbiota? I would assume there are microbiome data published from avians/pigeons. Please discuss your hypothesis for Cg as a gut microbia a bit more on the basis of recent findings of avian microbiome data (e.g. doi: 10.3389/fmicb.2020.01789. eCollection 2020; doi: 10.1371/journal.pone.0217804. eCollection 2019). A deeper discussion in this context is necessary in order to hold the title of the review "...for the avian microbiota-gut-brain axis"

Reviewer 2 Report

In this manuscript the author Yokoshita Nakayama reviews the roles and mechanistic of the mechanosensitive channels in Corynebacterium glutamicum for osmoregulation and L-glutamate secretion. The Gram-positive C. glutamicum is an industrial workhorse for the large-scale production of amino acids (especially L-glutamate) and posses a cell-wall structure similar to Mycobacteria. Therefore C. glutamicum is in interesting model organism for many aspects. Two mechanosensitive channels (MscCG and MscCG2) have been identified from different C. glutamicum islolates, which act in addition to their role for osmoregulation also as specific glutamate exporters. This additional function of these two mechanosensitive channels is interesting for both the general understanding of channels/transport as well as the application for glutamate production. The very high interest in this topic is also reflected by recently published reviews of several aspects of this field e.g. Kawasaki, H., & Martinac, B. (2020). Mechanosensitive channels of Corynebacterium glutamicum functioning as exporters of l-glutamate and other valuable metabolites. Current Opinion in Chemical Biology, 59, 77-83. Thus, the new aspect “avian microbiota-gut-brain axis” mentioned in the title and put into focus also in the abstract of this manuscript becomes very interesting for the reader.

In the manuscript the author first reviews the role of mechanosensitive channels for osmoregulation, then describes the specific properties of MscCG for glutamate export, and then focuses in the next two sections on mechanosensing and the triggers causing glutamate excretion. In the final section the author discusses potential roles for glutamate export by C. glutamicum in the scenario of being a soil or a gut bacterium.

The manuscript is interesting to read and contains good figures (especially Figure 5, which summarizes all the different triggers for glutamate excretion). However, despite being put into the focus of the reader’s attention in the heading and abstract the section on the possible role of C. glutamicum for avian microbiota-gut axis is rather short and superficially and requires more comprehensive literature work to achieve a solid basis for discussion.

Major concern:

The discussion on the physiological significance of the L-glutamate production for Corynebacterium glutamicum is missing any evidence. For its role as a soli bacterium and interaction with plants no reference is cited. At least a thorough search for indications in e.g. the genome repertoire of C. glutamicum required for interactions with plants or any data from plant-microbiome studies for presence of C. glutamicum. Also the hypothesis that C. glutamicum is part of the avaian microbiome is based on the detection of Actinobacteria in one cited microbiome study. On the one hand actinobacteria isolated from gut/feces samples include Bifidobacteria, which are well known to compromise a large part of the microbiome of wild animals and birds and are equipped with several features for colonization, thus the detection of actinobacteria is not indicative for presence of C. glutamicum or close relatives. Moreover, other studies even discuss the detection of Corynebacterium species as a potential indication for contamination (Hird, Sarah M., et al. "Comparative gut microbiota of 59 neotropical bird species." Frontiers in microbiology 6 (2015): 1403.). As also C. glutamicum has been isolated several times (including samples probably not from soils with expected high amounts of bird feces), I think this part of the manuscript is a very interesting, new aspect making a difference to other manuscript dealing with glutamate excretion, but it needs very careful revision.

Minor concerns:

  • Throughout the text – Corynebacterium glutamicum can be shortened to C. glutamicum
  • Page 1: “ Corynebacterium glutamicum was first isolated as a glutamate producer in the avian…” It needs to be specified that this was strain ATCC 13032 – other glutamicum strains were isolated from different materials – and as these isolates also differ in the presence of e.g. MscCG2 I suggest to specify the strains thoughout the text.
  • Page 2 – “Msccg2 as a minor C-glutamate exporter”: The experimental results of properties of MscCg2 with MscCG should be explained in the text.
  • Page 3, Fig.1: Maybe MscCgL should be included in the figure (but not as glutamate exporter!)
  • Page 3, Figure caption 1: A clear statement of the distribution of MscG2 among different C. glutamicum strains should be included in the text.
  • Page 3: “Corynebacterium glutamicum was discovered in avian feces-contaminated soil as a glutamate producer” – this is the case for ATCC13032, please specify.
  • Page 4: “This evidences that the corynebacterial mechanosensitive solute efflux system is not required for survival upon hypoosmotic downshock.” The paper from Nottebrock et al., was published before all mechanosensitive channels of E. coli were described – can it be excluded, that C. glutamicum does not posses a further, unknown mechanosensitive channel? – otherwise the data from Nottebrock et al., are no evidence that the solute eflux is not required.
  • Page 6: Change ”Backer et al.” to ”Becker et al.”
  • Pages 6 to / and figure 3A – why is the organization of MscCg2 not included/discussed?
  • Page 8: “For corynebacterial mechanosensing by MscCG-type mechanosensitive channels, the cell wall may function as extracellular molecular tethers to transmit the mechanical force to the channels by connecting with the extracellular loop of MscCG channels since the gain-of-function mutations that cause the spontaneous L-glutamate secretion are localized at the extracellular boundary (66).” Is this conclusion also justified for MscCG2, which lacks large parts? Are amino acid-residues conserved between MscCG and MscCG2?
  • References: The references need very thorough revision: Many references are incomplete (issue and page information lacking) or wrongly formatted.
  • Page 15: Conflicts of Interest: "The authors declare no conflict of interest." shoule be changed to "The author declares no...."

Round 2

Reviewer 2 Report

The manuscript was carefully revised, the suggestions and comments of the reviewers have been properly addressed by the author.